# Insulitis in Human Type 1 Diabetic Pancreas: From Stem Cell Grafting to Islet Organoids for a Successful Cell-Based Therapy

**DOI:** 10.3390/cells11233941

**Published:** 2022-12-06

**Authors:** Marcella La Noce, Giovanni Francesco Nicoletti, Gianpaolo Papaccio, Vitale Del Vecchio, Federica Papaccio

**Affiliations:** 1Department of Experimental Medicine, University of Campania “L. Vanvitelli”, Via L. Armanni 5, 80138 Naples, Italy; 2Multidisciplinary Department of Medical-Surgical and Dental Specialties, University of Campania “L. Vanvitelli”, Via L. de Crecchio 6, 80138 Naples, Italy; 3Department of Medicine, Surgery and Dentistry “Scuola Medica Salernitana”, University of Salerno, Via Salvador Allende, 84081 Baronissi, Italy

**Keywords:** beta cell, insulitis, stem cells, cell-based therapy, organoids, type 1 diabetes

## Abstract

Type 1 diabetes (T1D) is an autoimmune disease with immune cells’ islet infiltration (called “insulitis”), which leads to beta cell loss. Despite being the critical element of T1D occurrence and pathogenesis, insulitis is often present in a limited percentage of islets, also at diagnosis. Therefore, it is needed to define reproducible methods to detect insulitis and beta-cell decline, to allow accurate and early diagnosis and to monitor therapy. However, this goal is still far due to the morphological aspect of islet microvasculature, which is rather dense and rich, and is considerably rearranged during insulitis. More studies on microvasculature are required to understand if contrast-enhanced ultrasound sonography measurements of pancreatic blood-flow dynamics may provide a clinically deployable predictive marker to predict disease progression and therapeutic reversal in pre-symptomatic T1D patients. Therefore, it is needed to clarify the relation between insulitis and the dynamics of β cell loss and with coexisting mechanisms of dysfunction, according to clinical stage, as well as the micro vessels’ dynamics and microvasculature reorganization. Moreover, the ideal cell-based therapy of T1D should start from an early diagnosis allowing a sufficient isolation of specific Procr+ progenitors, followed by the generation and expansion of islet organoids, which could be transplanted coupled to an immune-regulatory therapy which will permit the maintenance of pancreatic islets and an effective and long-lasting insulitis reversal.

## 1. Introduction

Type 1 diabetes (T1D) is an autoimmune disease characterised by immune cells infiltration of Langerhans islets, leading to an inflammatory state called “insulitis” [1]. It affects capillary vessels and impairs the related blood flow. The result is a severe loss of pancreatic beta-cells, whose debris are phagocyted by macrophages. This inflammatory lesion, consisting of immune cells (mainly autoreactive T-lymphocytes) located around and within the islets, is considered by several authors as a key histopathological feature, although not investigated in all its aspects. 

The infiltrating immune cells reach the islets through blood and lymphatic vessels as well as through extra-cellular matrix (ECM) spaces. During the last decades, new methodologies advanced our understanding on islet physio-pathology and identified the predominant cellular types that infiltrate the islets; moreover, molecular processes associated with insulitis have been studied [1]. While insulitis is the critical element of T1D occurrence and pathogenesis, it is often present in a limited percentage of pancreatic islets, even at diagnosis, with variable relation to disease duration. In fact, insulitis and beta-cell decline occur years before diabetes clinical signs onset, leading to poor diagnosis or delayed patient’s care. Therefore, it is required to achieve a non-invasive detection of insulitis and beta-cell decline, allowing the correct and early diagnosis and to monitor therapy. However, this goal is still not reachable for several reasons, starting from the morphological islet microvasculature, that is rather dense and rich, as for all endocrine glands. In addition, micro vessels are considerably rearranged during diabetic insulitis. The latter is of key importance to monitor the progression of the disease and must be studied in deep along with insulitis steps and its unpredictable occurrence.

Pancreatic islet microcirculation exhibits distinctive features, with an islet capillary network showing five times higher density than the capillary network of the exocrine counterpart and high permeability. Moreover, the islet microvascular endothelial cells show about 10 times more fenestrations than those of the exocrine tissue. In an interdependent physical and functional relationship with beta cells, islet endothelial cells are involved not only in the delivery of oxygen and nutrients to endocrine cells, but they also induce insulin gene expression during islet development, affect adult beta cell function, promote beta cell proliferation, and produce several vasoactive, angiogenic substances and growth factors. Specific markers of islet microvasculature are alpha-1 proteinase inhibitor and nephrin, a highly specific barrier protein with adhesion and signalling function. The islet micro endothelium also appears to have a role in fine-tuning blood glucose sensing and regulation, and to behave as an active “gatekeeper” in the control of leukocyte recruitment into the islets, adopting an activated phenotype during autoimmune insulitis in T1D. Therefore, this dense vasculature possibly plays a role in the physiology as well as in the disease of the pancreatic islets. In this review we will describe the phenotypic and functional characteristics of islet micro endothelium and its possible involvement in type 1 and 2 diabetes, and islet revascularization in transplantation setting [2]. 

Recently, contrast-enhanced ultrasound sonography (CEUS) measurements of pancreatic blood-flow dynamics have been proposed as a non-invasive tool to predict disease progression in T1D pre-clinical models. Authors have demonstrated that streptozotocin-treated NOD and adoptive-transfer mice had altered islet blood-flow dynamics prior to diabetes onset, consistent with islet microvasculature reorganization. These assessments predict both time to diabetes onset and future responders to anti CD4-mediated disease prevention. Thus, CEUS of pancreas blood-flow dynamics could provide a clinically predictive technology for disease progression monitoring in pre-symptomatic T1D and eventually counteract it, avoiding disease onset [3].

In addition, as previously evidenced, the islet infiltration can occur also through the ECM spaces, where the immune cells accumulate. During this process micro vessels are strongly rearranged. They at first allow immune infiltrating cells to extravasate and then, when the infiltrate enlarges, they undergo a complete rearrangement [4].

Therefore, it remains a major goal to clarify the relation of insulitis with the dynamics of beta cell loss and coexisting mechanisms of dysfunction, according to clinical stage, as well as the micro vessels’ dynamics and microvasculature reorganization.

The key to design therapeutic strategies that target multiple mechanisms is to improve the understanding of the insulitis pathogenesis.

Immune-cell infiltration into the islets of Langerhans (insulitis) and beta-cell failure occur years before T1D clinical onset. Non-invasive detection of insulitis and beta-cell loss would allow an early diagnosis of diabetes and provide a way to monitor therapeutic intervention. However, clinical approaches for specific and non-invasive imaging disease progression, in this setting, are not yet available. 

Islets, being endocrine glands, present a dense microvasculature that is strongly reorganized during diabetes, due to the presence of infiltrating cells inside and around the islets and vessels and concurrent beta cell decrease. Therefore, the islets are progressively altered in their structure, resulting massively inflamed. It is evident that the islet microvasculature starts to be reorganized much earlier than beta cell loss occurrence, as the inflammatory cells reach the peri-islet and islet blood vessels to extravasate [5].

Years ago [4,6,7] we described the relationship between insulitis and islet microvasculature. We focused on the histophysiology of the islets’ microvasculature showing that islet micro vessels undergo a substantial alteration with strong reduction and remodelling and this, in turn, affects the physiological blood flow, taking into consideration that beta cells are firstly perfused and influence both endocrine non-beta islet and peri-insular exocrine cells. 

Moreover, islets’ vascular alterations, which mainly occurr at the level of post-capillary venules encircling the islets of Langerhans, was concomitant to a fall in superoxide-dismutase (SOD) activity [8]. We argued that these findings, together with the increase in vascular permeability and the morphological evidence of areas of oedema formation within the islets, raised the interest to the insulitis and blood flow alterations, confirming that the islet vascular system is involved in early insulitis and contribute to beta-cell lysis [9]. 

## 2. Insulitis Characterisation

T1D is regarded as a T-cell-mediated autoimmune disorder, and there are many lines of experimental, clinical, and pathological evidence supporting this view [1]. The pathologic hallmark of T1D has long been considered the inflammatory lesion of the pancreatic islets (insulitis), characterised, as previously explained, by the presence of immune and inflammatory cells within and around the pancreatic islets [7]. Insulitis, firstly described by Gepts [10] is the appearance of the autoimmune attack against beta cells. Other studies, as well as those focusing on other features of pancreas pathology in T1D, are very limited [11,12,13,14].

Campbell defined insulitis as an islet with six or more CD3+ cells immediately adjacent to or within the islet with three or more islets per pancreas section, and classified the insulitis lesion according to the following features: (i) presence of a predominantly lymphocytic infiltration of the islets of Langerhans; (ii) detection of at least #15 CD45+ cells/islet and (iii) presence of a lesion in a minimum of three islets. [15]. Moreover, the number of CD3+ cells/islet varied in each donor, irrespectively of islet insulin immunopositivity or size, and CD3+ cells were observed to diffusively infiltrate the islets, in aggregates of various size, or both [14].

Insulitis, as previously pointed out, can be found either in the islet periphery (peri-insulitis, where inflammatory cells are at one pole of the islet and in contact with the islet periphery) or within the islet parenchyma (intra-insulitis); most studies indicate that the predominant form of insulitis in the human pancreas is the “peri-insulitis”. It has been reported as to be much less severe than in experimental mouse models [16,17]. The latter is doubtful and maybe is only due to the longer time of the process. 

Moreover, it has been reported that insulitis is often detected in islets containing insulin-positive beta cells, but the same pancreas may contain also atrophic islets devoid of insulin-positive cells: the latter may be due either to a previous insulitis or to a direct action on islets’ beta-cells by autoantibodies [15]. Therefore, the process has a lot of variables.

Studies found that heterogeneous profiles are linked to disease severity and progression. While both T and B lymphocytes are reported in insulitis lesions, cytotoxic CD8 T-cells appear to be the predominant population and could target beta cells expressing elevated levels of Human Leucocyte Antigen (HLA) class I molecules [18]; moreover, over-expression of class I (and class II) molecules may be associated with viral infections, suggesting a key role in T1D pathogenesis [19]. Over-expression of HLA molecules represents another feature of T1D pathogenesis that emphasises a chronic inflammatory condition, associated with insulitis [20].

It has been postulated that the presence of higher proportions of B-cells in the insulitis lesion may be a marker of early triggering of autoimmunity or of a more rapid rate of beta cell loss.

Insulitis involves a wide array of cell types—T and B lymphocytes as well as myeloid cells—macrophages and dendritic cells [21], which can take on the organization of typical tertiary lymphoid structures [22,23].

The proportion of infiltrated islets and the extent of infiltration appear generally lower in human patients than in NOD mice, and a dominance of CD8+ over CD4+ T cells seems frequent [24,25], with a variable frequency of B lymphocytes [26].

Leete et al. reported that in very young children (especially those <7 years at onset) pancreatic islets are infiltrated by both CD8+ T- and CD20+ B-lymphocytes in roughly equal proportions, but it is widely held that the CD8+ T-lymphocytes are responsible for driving beta-cell toxicity [27]. By contrast, the role played by B-lymphocytes remains unclear. This is compounded by the fact that in older children and teenagers (those ≥13 years at diagnosis) the proportion of B-lymphocytes found in association with inflamed islets is much reduced in comparison with those who are younger at diagnosis (reflecting two types of disease) whereas CD8+ T-lymphocytes constitute the predominant population in both groups [27].

Leete et al., in 2016, found that the proportion of B-lymphocytes present within the islet infiltrate varies according to the stage of islet destruction. Moreover, it also varies in parallel with the number of CD8+ T-cells but not in proportion to the numbers of CD4+ cells. In all cases under study, it was clear that CD8+ T-cells are the predominant population as insulitis develops. They also noted that the influx of B-lymphocytes closely mirrors the CD8+ cell profile. 

The B-lymphocytes present in and around the inflamed islets are not engaged in autoantibody production, as they did not lose the surface marker CD20.

Based on the proposition that CD20+ B-lymphocytes might play a role in promoting the activation state of CD8+ T-cells in and around inflamed islets, islet-associated B-lymphocytes may play a profound role in influencing the outcome of autoimmunity in T1D, perhaps by regulating the cytotoxic activity of their CD8+ T-cell counterparts [28].

Some key molecular changes, typical of insulitis, have been detected: the accumulation of hyaluronan, a key constituent of the ECM, and hyaluronan binding proteins, around islet cells and infiltrating lymphocytes in islets affected by insulitis [29]. These molecules may play a role in insulitis by promoting lymphocyte adhesion and migration. In addition, antagonizing hyaluronan deposition prevents diabetes in mice [30,31].

Therefore, the role of the ECM in insulitis pathogenesis is important and more than expected being either a peri- and intra-islet infiltration, having relationships with the connective tissue internal and surrounding the islet.

## 3. Insulitis in Human Type 1 Diabetes

As highlighted before, insulitis in human T1D pancreas can be detected in a low percentage of cases and up to 30% [1]. The variability depends on age and disease duration. Insulitis is predominant in younger patients and when they are tested near diagnosis [16]. The frequency of insulitis may display limited inverse correlation with diabetes duration and has no correlation with age, it predominantly affects insulin-positive islets and has been also observed in patients with residual beta cells, also years after diagnosis [14].

The results obtained analysing serial biopsies up to now highlight the chronicity of this process, detectable for years after diagnosis, either in children or young adults. 

Moreover, insulitis does not affect all the islets at the same time, confirming that this is a long evolving process.

Of relevance, there was no correlation of beta cell mass with insulitis, disease duration and age of onset. Inflammation and beta cell dysfunction may be an important pathogenic mechanism at the time of initial disease manifestation and contribute to cause the symptoms of severe hyperglycaemia [32].

All those findings suggest that beta cell destruction is quite heterogeneous and is not likely to be completed until several years after diagnosis. Studies have reported persistence of insulin-positive beta cells even decades after diagnosis [33] and that glucose transporters continue to be expressed [34]. In some patients with long disease duration, beta cells express the survivin molecule, possibly a factor involved in the persistence of the beta cell phenotype [35].

Low levels of beta cell apoptosis have been noted in the pancreas of patients with long disease duration, implying the existence of some beta cell turnover [33].

The above observations are usually present in the context of chronic signs of islet inflammation, including insulitis and increased expression of HLA class I molecules [36].

Therefore, this kind of disease onset does not help either an early diagnosis or therapeutic design. On the other hand, it would also be important to prevent the disease in some cases when familiarity is present.

Recent data have highlighted that a Slowly Progressive type 1 insulin-dependent Diabetes Mellitus (SPIDDM), sometimes indicated to as a Latent Autoimmune Diabetes in Adults (LADA), is a heterogeneous disease often confused with type 1 and type 2 diabetes. In the pancreas of patients with SPIDDM, which includes a T-cell-mediated insulitis, the following features can be detected: pseudo-atrophic islets, absence of beta cells, and the lack of amylin (i.e., islet amyloid polypeptide) deposition into the islet cells (a pathologic marker of type 2 diabetes). Being therefore this a T1D syndrome with a slow evolution, the authors claim for prevention strategies for patients with SPIDDM. The proposed strategies aim at the preservation of beta-cell function and include the early administration of low doses of insulin or of dipeptidyl peptidase-4 (DPP-4) inhibitors [37].

Recently, beta cell autophagy has been hypothesized to be involved in T1D pathogenesis [38]. This is in line with the knowledge that T1D is a multifactorial autoimmune disease involving multiple environmental and genetic factors, but still without a clear aetiology. Early signalling defects within the beta cells may promote a change in the local immune milieu leading to autoimmunity. Therefore, some studies have been focused on intrinsic beta-cell mechanisms that aid in the restoration of cellular homeostasis under environmental conditions that cause dysfunction. One of these intrinsic mechanisms to promote homeostasis is autophagy (linked with beta-cell dysfunction in type 2 diabetes).

## 4. Attempts to Counteract or Prevent Insulitis

The islet of Langerhans has been defined “a remarkably sophisticated micro-organ” and this reflects the great problems in regenerating it, starting from beta cells. In some studies, reactive oxygen species (ROS) were found to be involved in insulitis pathogenesis, where the “vasculitis” could be imagined as a systemic process [6], albeit islet beta cells are the only cells completely damaged. Several attempts have been proposed to prevent or counteract insulitis and T1D up to now. Most of the proposed therapies were studied in animals, namely STZ-treated C57BL6J mice in low doses [39] and NOD mice [40].

Over the decades a huge number of studies aimed to find new insights into T1D treatment, to change the therapeutic approach, moving from the treatment of the insulitis to the recovery of the damaged tissue, with very poor results. Specifically, recent and ongoing clinical trials are based on the use of Non-Antigenic (NAg) drugs, targeted therapy, and Antigen (Ag) specific agents for the T1D treatment.

Formerly, cyclosporine has been a widely adopted as a NAg drug with high efficacy in preserving the beta-cell function in new T1D diagnosed patients [41,42]. Many other NAg drugs have been employed for the T1D treatment during the years, among which: Azathioprine [43], histamine agonists Ketotifen [44] and Nonicotinamide [45] as monotherapy or Anti-Thymocyte Globulin (ATG) in combination with Prednisone [46]. Many of the reported NAg drugs showed a significant short-term efficacy, but none of these could preserve the beta cells function or eventually prevent T1D onset, sometimes also with severe side effects.

The innovative approach in T1D patient treatments, based on the precision medicine definition, allowed to set up new targeted therapy agents. Promising results in reducing the immunogenicity against the islets of Langerhans have been achieved with the use of the human anti-CD3 monoclonal antibody (mAb) [47], even if they were not able to affect the early autoimmune process. Nevertheless, a recent midterm clinical trial on the second generation anti-CD3 mAb Otelixizumab demonstrated a reduced exogenous insulin request in early diagnosed T1D patient [48]. Although further promising results have been obtained also with Teplizumab, another anti CD-3 mAb which showed promising results in T1D treatment due to its ability to counteract the C-peptide levels reduction in a subset of patients with low HbA1c [49], its effectiveness was not demonstrated in phase III trials [50,51]. Furthermore, the anti-CD20 mAb Rituximab demonstrated a higher islet beta-cell function preservation after one year in T1D patients [52], which completely reverts after 30 months, at the same time exposing patients to a higher susceptibility to chronic infections [53]. Other immune-modulators, such as Abatacept, a Cytotoxic T Lymphocyte antigen-4 (CTLA-4) immunoglobulin fusion protein, and Anakinra or Canakinumab, interleukin-1 (IL-1) inhibitors, showed high ability in T cells activation blocking, even in this case only at short term [54].

Most of the described NAg immunomodulatory agents have mainly been tested to delay the disease progression, instead of focusing on the extensive beta-cell destruction and the relative insulin deficiency. Pioneering studies identified insulin as a very advanced autoantigen in insulitis [55]; unfortunately, even though many favourable in vivo pre-clinical studies encouraged the adoption of insulin and/or its mimotopes, to shut off the autoimmune response [56], clinical trials failed because the subcutaneous administration of active insulin, needed for the immunization, dramatically increased the risk of hypoglycaemia [57]. More recent studies demonstrated in T1D patients with high titre of autoantibodies anti-insulin subjected to oral insulin administration, a delay in the disease onset [58]. Considering the described side effects provoked by the exogenous insulin administration, many scientists turned on the use of safer antigens, such as the DiaPep277, a peptide derived from the Heat shock protein 60 (Hsp60) with a strong immunomodulatory potential. DiaPep277 showed a strong effect in protecting NOD mice from diabetes [59], with very interesting results also in phase III clinical studies [60].

Of interest, substantial deposition of the ECM component hyaluronan (HA) is characteristic of insulitis type 1 diabetes. Recently, it has been investigated whether HA accumulation may be early detectable in islets in disease pathogenesis and how this may affect the development of insulitis and beta cell mass. Bogdani et al. have revealed that islet HA accumulation is a rather early event in the pathogenesis of autoimmune diabetes [61]. These findings, consistent in humans with early islet autoimmunity, support a crucial role exerted by HA in promoting islet inflammation in type 1 diabetes as previously supposed by the same authors [62]. It may be worthy to explore the mechanisms regulating HA mass in islets and the interactions between HA and islet cells or immune cells, to develop therapeutic interventions targeting HA accumulation which, in turn, may stop the development of insulitis.

The latter seems to be a suitable physio pathological approach that can lead to positive results.

## 5. Stem Cell Grafting

Insulitis and the related micro vessel alterations remain a critical element of T1D pathology, having a great relevance to design more effective targeting of pathogenic mechanisms.

It is well known that combination therapies promoting immune regulation and addressing beta cell dysfunction should be more effective in treating this chronic disease, but studies should be much more focused on the phenotypic features of infiltrating cells in the insulitis lesion, also in relation to other abnormalities, as well as gene expression profile and epigenetic regulation of the beta cells, which could be a major contributor to their dysfunction and death.

More recently the use of stem cell grafting has been proposed. Clinical good-manufacturing practice (cGMP)-grade stem cell products have been used in human clinical trials, showing that stem cell transplantation has beneficial effects on T1D, with no obvious adverse reactions. To better achieve remission of T1D by stem cell transplantation, innovative approaches such as mesenchymal stem cells (MSCs), human embryonic stem cells (hESCs), and bone marrow hematopoietic stem cells (BM-HSCs) to restore the immunotolerance and preserve the islet beta-cell function of T1D have been investigated [63], but the results are, up to now, discouraging.

In fact, studies lacked a good base of pre-clinical investigations attempting endocrine pancreas recovery using stem cell transplantation. The focus was on the use of hematopoietic and mesenchymal stem cells (MSCs) [64]. However, it should be considered that MSCs could exert an anti-inflammatory role, thanks to the secretion of chemokines and cytokines capable of immunomodulation and T cell inhibition, with the potential of improving strategies of engraftment of donor islets [65].

Authors tried to use all types of stem cells with partial, incomplete, or largely negative results, mainly because the goal to overcome the disease is far from being obtained. Stem cells can only replace for a given time the lost cells, but they will inevitably convert to new targets of the immune attack. Sometimes they can have more time or better endure the insults, but the true resistant ones are far from being obtained.

Therefore, more effective therapeutic approaches are needed [66].

To overcome these issues, combinatorial strategies have been proposed for a curative treatment of T1D [67]. Combining safe and effective stem cell strategies with reliable existing therapies such as islet transplantation, as well as the immunosuppressive and immunomodulatory drug regimens and/or novel bioengineering techniques and/or gene therapies, would ensure an optimistic scenario for successful translation of stem cell therapy in the cure of T1D.

In our opinion, every strategy including stem cells of every origin albeit differentiated into insulin-containing cells or whole islets for a mere grafting doom to failure, as the history of the previous studies teaches. This since the grafted cells or islets will be inevitably de novo attacked, unless other strategies are undertaking, for instance by encapsulation [68] or gene editing approaches that protect them from the immune attack [69]. Indeed, encouraging data come from the 20-year outcomes of a cohort study employing islet transplantation [70].

## 6. Cell Therapy and the Organoid Perspective

Cell therapy, instead of a mere stem cell grafting, provides an opportunity to prevent or reverse T1D [71]. 

In fact, the clinical trial results of stem cell therapies for T1D are largely dissatisfactory [72], and many questions and technical hurdles still need to be solved. The major points that researchers should overcome include: (i) how to generate more mature functional beta cells in vitro from hPSCs; (ii) how to improve the differentiation efficiency of IPCs from hPSCs; (iii) how to protect implanted IPCs from autoimmune attack; (iv) how to generate enough desired cell types for clinical transplantation and (v) how to establish thorough insulin independence [73,74]. 

However, they seem to not take into consideration that it is needed to solve the main problem at the basis of the disease. Also, in this case it seems that effectively diabetes stem cell therapy takes evasive actions [75].

Therefore, the application of cell-based therapy for T1D albeit represents the most advanced approach for curing it, must take into consideration also the main variables. 

Firstly, protective encapsulating devices and gene-editing technologies could obviate the need for antirejection drugs in stem-cell-derived therapies for diabetes.

Then, adoptive transfer of autologous cells having enhanced immunomodulatory properties could suppress autoimmunity and preserve beta-cells. Such therapies have been made possible by a combination of genome-editing techniques and transplantation of tolerogenic cells. In-vitro modified autologous hematopoietic stem cells and tolerogenic dendritic cells may protect endogenous and newly generated beta-cells without hampering immune surveillance for infectious agents and malignant cellular transformation. 

The technological advances will allow to test the proposed new strategies directly in human cell models, as the gold standard in biomedical research fields. Omics approaches to characterize the molecular signatures of the better cell populations, should be given priority to accelerate bioengineering strategies. This will help to decipher the still mysterious treasure of human islet beta-cell regeneration, as well as it will prevent the recurrent drawbacks in the later stages of clinical testing or the overlooking of potentially effective treatments that fail in mouse model testing.

However, methods to generate cells that meet quality and safety standards for clinical applications have been recently addressed [74], despite they are still far from application in clinical setting. 

On the other hand, a vast number of studies that have been reported on gene therapy for the management of T1D have been conducted in animal models and in preclinical studies. Some recent reports from phase I/II clinical trials showed the feasibility of such treatments in humans, although with a small *n* [76]. Currently, there are several gene level interventions that are being investigated, notably overexpression of genes and proteins, transplantation of cells that express the genes against T1D, stem-cells mediated gene therapy, genetic vaccination, immunological precursor cell-mediated gene therapy and vectors. The way to better address these issues has been indicated [77].

Anyway, cell therapy is a much more reliable perspective and studies must be performed to this aim.

The latest research interest for either deeper knowledge or possible therapeutic advances are pancreatic islet organoids.

It is well known how hard it is to produce functional beta cells in vitro. Recently [78], a previously unidentified protein C receptor positive (Procr+) cell population has been identified in adult mouse pancreas through single-cell RNA sequencing (scRNA-seq). The cells are present inside the islets, do not express differentiation markers, and show epithelial-to-mesenchymal transition characteristics. By means of genetic lineage tracing, it has been demonstrated that Procr+ islet cells undergo clonal expansion and generate all four endocrine islet cell types during adult homeostasis. Sorted Procr+ cells, representing ∼1% of islet cells, have been demonstrated to be capable to form islet-like organoids when cultured at clonal density. In addition, exponential expansion can be maintained over long periods by serial passaging, while differentiation can be induced at any time point in culture. Interestingly, beta cells dominate in differentiated islet organoids, while α, δ, and PP cells occur at much lower frequencies. The organoids are glucose-responsive and insulin-secreting. Of great scientific significance, upon transplantation in diabetic mice, these organoids have been found to reverse the disease. Therefore, the adult mouse pancreatic islet contains a population of Procr+ endocrine progenitors that can be the more effective cells to be used for replacement also in T1D [78]. This approach, if confirmed in humans, could represent an interesting source of human islets. 

In conclusion, one of the proposed approaches for cell-based therapy of T1D in the future (Figure 1) could combine an early diagnosis (possibly thanks to CEUS) with a sufficient isolation of Procr+ progenitors, followed by the generation and expansion of islet organoids, which could be transplanted coupled to an immune-regulatory therapy which will permit the maintenance of pancreatic islets and an effective and long-lasting insulitis reversal. A key issue to resolve would be the maximization of progenitors’ isolation and organoids expansion, such that only a few biopsies would ensure an optimal outcome.

## Figures and Tables

**Figure 1 cells-11-03941-f001:**
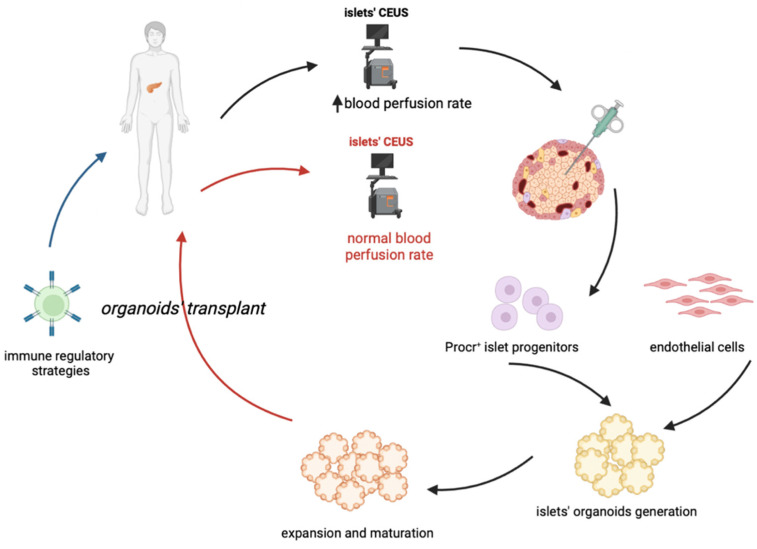
Proposed cell-based treatment of type 1 diabetes. The proposed approach should combine an early diagnosis of T1D, through CEUS (Contrast-Enhanced Ultrasonography) with the proper quantity of Procr+cell progenitors, combined with endothelial cells, derived from IPSCs. This in order to generate and expand islet organoids, that can be then grafted in patients undergoing an immune-regolatory therapy. The latter is required to guarantee a long-term maintenance of the transplanted pancreatic islets.

## Data Availability

Not applicable.

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
