# Peer review of "Insulitis in Human Type 1 Diabetic Pancreas: From Stem Cell Grafting to Islet Organoids for a Successful Cell-Based Therapy"

_cells, 2022, doi:10.3390/cells11233941_

Round 1

Reviewer 1 Report

Noce et.al describes a critical topic in the field of T1D development and the future direction of cell-based therapy in combating the disease. The modified version is well-structured and the literature is up-to-date. In my view, this paper will be sufficient for publication after some language polish which is suggested to be revised by native speakers.

Author Response

We thank the reviewers for their valuable comments. We provide a point-by-point response. Changes in the manuscript are tracked in red.

Noce et.al describes a critical topic in the field of T1D development and the future direction of cell-based therapy in combating the disease. The modified version is well-structured and the literature is up-to-date. In my view, this paper will be sufficient for publication after some language polish which is suggested to be revised by native speakers.

We thank this reviewer for his positive comment. The manuscript has been checked by a native English speaker.

Reviewer 2 Report

Overall, this paper focuses on the effect of insulitis in the development of T1D and in the requirement of early detection with stem cell grafting and islet differentiation being a minor part of the paper. Furthermore, the use of stem cells in diabetes and islet differentiation is not explained in detail or appropriately referenced. This section would need to be extended to fit the title of the manuscript. Different ideas throughout the text are not well linked and the overall text would benefit from revision by a native English speaker.

Abstract

·       Line 17-19. Sentence is not clear. It is therefore required to establish non-invasive detection markers for insulitis and β-cell decline to allow accurate and early diagnosis and to provide means to monitor therapy.

·       Line 19 to 21. Merge both sentences. However, this goal is still not reachable due to the morphological islet microvasculature, which is rather dense and rich, and is considerably rearranged during diabetic insulitis.

·       Line 22 to 25. Rearrange sentence. Further microvascular studies are required to understand if contrast-enhanced ultrasound sonography measurements of pancreas blood-flow dynamics may provide a clinically deployable predictive marker for disease progression in pre-symptomatic T1D and therapeutic reversal.

Significance statement

·       Line 42. The authors state that stem cell-based therapies are not directed against the cause but the consequences and suggest that organoids are promising for this matter. However, organoids, or stem cell islets, are stem cell-based therapies. Rephrase.

1. Introduction.

·       Line 47. Change involves from affects.

·       Line 60: beta or β. Pick one and stay consistent throughout manuscript.

·       Line 65: what does it refer to?

·       Line 88: STZ abbreviation is missing

·       Line 89: remove the word mice: STZ-treated NOD mice

·       Line 91 – 92: reference required

·       Line 94: clarify therapeutic reversal. Beta cell loss cannot be reversed

2. Insulitis characterization

·       Line 140: what is considered lesion

·       Line 171 – 174: reference is missing.

3. Insulitis in human T1D

·       Line 201 – 202: reference required

·       Line 226: it would also be important; modify.

·       Line 232 – 233: lack of amylin; modify.

·       Line 234 -236: what sort of prevention strategies could be taken?

4. Attempts to counteract or prevent insulitis

·       Line 246: Islet of Langerhans have been defined as; modify.

·       Line 300: ECM; modify

5. Stem cell grafting

·       Lines 332 – 332. It should be mentioned that HSCs and MSCs have been used because of their immunomodulatory potential as a result of chemokine and cytokine secretion (Reference PMID: 33865426) to improve the engraftment of donor islets or to reduce the inflammation.

·       Line 341-345: 20-year outcomes of islet transplantation have recently been published (PMID: 35588757) demonstrating that islet transplantation can cure diabetes. However, one of the major limitations of islet transplantation remain the lack of donor islets. For this reason, iPSCs and ESCs offer a potential solution to this lack of islets.

·       Line 346 – 349. Auto-immunity is definitely a possibility, however, there are several strategies under development, including encapsulation and genetic modification.

6. Cell therapy and the organoid perspective

·       Lines 354 to 359: references are required. Include: PMID: 36272021, PMID: 36001981,

·       Lines 381- 382. Recent publication by Cuesta-Gomez et al has established the critical quality control and release criteria for clinical implementation of stem cell islets (PMID: 36001981).

·       Line 385: ViaCyte and Vertex clinical trials should be mentioned.

·       Lines 395 to 410. How could this be implemented into the clinic? Are the authors suggesting isolating this population from human islet donors and expand it to overcome the lack of islet donors?

·       Lines 411 to 415: are the authors suggesting partial or total removal of the pancreas from the T1D patient during early diagnosis to isolate Procr+ cells? Generation of stem cell islets from peripheral blood would entail a reduced burden to the donor with the same auto-immunity complications.

Furthermore, it would first need to be determined whether Procr+ progenitors from T1D patients are able to differentiate with the same efficiency as those from healthy controls.

·       Line 417. Figure legend needs to be expanded to describe the proposed cell-based treatment and resolution of the image must be improved.

Author Response

Response to reviewer’s

We thank the reviewers for their valuable comments. We provide a point-by-point response. Changes in the manuscript are tracked in red.

Reviewer 2

Overall, this paper focuses on the effect of insulitis in the development of T1D and in the requirement of early detection with stem cell grafting and islet differentiation being a minor part of the paper. Furthermore, the use of stem cells in diabetes and islet differentiation is not explained in detail or appropriately referenced. This section would need to be extended to fit the title of the manuscript. Different ideas throughout the text are not well linked and the overall text would benefit from revision by a native English speaker. 

 We thank this referee for the suggestions. We have altered the manuscript in all parts according to his comments. Everything can be found in red color within the text.

Abstract

  • Line 17-19. Sentence is not clear. It is therefore required to establish non-invasive detection markers for insulitis and β-cell decline to allow accurate and early diagnosis and to provide means to monitor therapy. 

            “It is therefore needed to obtain a non-invasively insulitis and β-cell decline detection to allow correct and early diagnosis and to provide means to monitor therapy.”

            Change to

            “Therefore, it is required to define reproducible methods to detect insulitis and beta-cell decline to allow accurate and early diagnosis and to pro provide means to monitor therapy”

  • Line 19 to 21. Merge both sentences. However, this goal is still not reachable due to the morphological islet microvasculature, which is rather dense and rich, and is considerably rearranged during diabetic insulitis.

                  “However, this goal is still not reachable due to the morphological islet microvasculature, rather dense and rich. In addition, it is considerably rearranged during diabetic insulitis. The latter is of key importance in order to monitor the progression of the lesion.”

                  Change to

                  “However, this goal is still far due to the same morphological aspect of islet microvasculature, which is rather dense and rich, and is considerably rearranged during diabetic insulitis.”

  • Line 22 to 25. Rearrange sentence. Further microvascular studies are required to understand if contrast-enhanced ultrasound sonography measurements of pancreas blood-flow dynamics may provide a clinically deployable predictive marker for disease progression in pre-symptomatic T1D and therapeutic reversal. 

            “On the other hand, if contrast-enhanced ultrasound sonography measurements of pancreas blood-flow dynamics may provide a clinically deployable predictive marker for disease progression in pre-symptomatic T1D and therapeutic reversal, it must be further supported by microvasculature studies, allowing to understand the major problem.”

            Change to:

            “Further microvascular studies are required to understand if contrast-enhanced ultrasound sonography measurements of pancreatic blood-flow dynamics may provide a clinically deployable predictive marker for disease progression prediction in pre-symptomatic T1D and therapeutic reversal.”

Significance statement

  • Line 42. The authors state that stem cell-based therapies are not directed against the cause but the consequences and suggest that organoids are promising for this matter. However, organoids, or stem cell islets, are stem cell-based therapies. Rephrase.

                  “Unfortunately, they have been disappointing as they are not directed against the cause but on the consequences (insulitis). New hopes come from gene-editing and organoids.”

                  Change to:

                  “Unfortunately, they have been disappointing as they are not directed against the cause but on the consequences (insulitis). New hopes come from the application of gene-editing and islets-organoids, as long as these more efficient technologies are coupled with therapies aiming at controlling the immune attack.”

  1. Introduction.

  • Line 47. Change involves from affects.

                  “It involves capillary vessels”

                  Change to:

                  “It affects capillary vessels”      

  • Line 60: beta or β. Pick one and stay consistent throughout manuscript. 

                  “β-cell” change to “beta-cell”

  • Line 65: what does it refer to? 

                  “In addition, it is considerably rearranged during diabetic insulitis.”

                  Change to:

                  “In addition, micro vessels are considerably rearranged during diabetic insulitis.”

  • Line 88: STZ abbreviation is missing

                  “STZ” change to “streptozotocin (STZ)”

  • Line 89: remove the word mice: STZ-treated NOD mice

                  “STZ-treated mice, NOD mice” change to: “STZ-treated NOD mice”

  • Line 91 – 92: reference required

                  We moved ref 3 at the end of the paragraph.

  • Line 94: clarify therapeutic reversal. Beta cell loss cannot be reversed

                  “therapeutic reversal” change to “and eventually counteract it, avoiding disease onset”

  1. Insulitis characterization 

  • Line 140: what is considered lesion

                  “presence of an insulitis lesion”

  • Line 171 – 174: reference is missing.

            We added ref 27 call out

  1. Insulitis in human T1D

  • Line 201 – 202: reference required

                  We added ref 1 call out

  • Line 226: it would also be important; modify

                  “it would be also important” change to “it would also be important”

  • Line 232 – 233: lack of amylin; modify. 

            “lacking of amylin” change to “lack of amylin”

  • Line 234 -236: what sort of prevention strategies could be taken?

                  “Being therefore this a T1D syndrome with a slow evolution, the authors claim for prevention strategies, and clinical options for patients with SPIDDM [37].”

            Add the following: “The proposed prevention strategies are based on the preservation of beta-cell function and include the early administration of low doses of insulin or of dipeptidyl peptidase-4 (DPP-4) inhibitors [37].”

  1. Attempts to counteract or prevent insulitis

  • Line 246: Islet of Langerhans have been defined as; modify. 

                  “The islet of Langerhans, containing the β cells, has been defined” change to “The islet of Langerhans, containing the β cells, have been defined”

  • Line 300: ECM; modify

                  “extracellular matrix” change to ECM

  1. Stem cell grafting

  • Lines 332 – 332. It should be mentioned that HSCs and MSCs have been used because of their immunomodulatory potential as a result of chemokine and cytokine secretion (Reference PMID: 33865426) to improve the engraftment of donor islets or to reduce the inflammation.

“In fact, studies lacked a good base of pre-clinical investigations attempting endocrine pancreas recovery using stem cell transplantation. The focus was on the use of hematopoietic and mesenchymal stem cells [64].” Change to “In fact, studies lacked a good base of pre-clinical investigations attempting endocrine pancreas recovery using stem cell transplantation. The focus was on the use of hematopoietic and mesenchymal stem cells (MSCs) [64].”

Add the following: “However, it should be considered that MSCs could exert an anti-inflammatory role, thanks to the secretion of chemokines and cytokines capable of immunomodulation and T cell inhibition (doi: 10.1186/s12967-021-02822-5), with the potential of improving strategies of engraftment of donor islets.”

  • Line 341-345: 20-year outcomes of islet transplantation have recently been published (PMID: 35588757) demonstrating that islet transplantation can cure diabetes. However, one of the major limitations of islet transplantation remain the lack of donor islets. For this reason, iPSCs and ESCs offer a potential solution to this lack of islets. 

We agree with the reviewer that this important study must be cites as it is relevant in the field. However, we would be more cautious in affirming that this approach can cure rather than control diabetes.

We added the following sentence at the end of the paragraph: “Indeed, encouraging data come from the 20-year outcomes of a cohort study employing islet transplantation (doi: 10.1016/S2213-8587(22)00114-0).”

  • Line 346 – 349. Auto-immunity is definitely a possibility, however, there are several strategies under development, including encapsulation and genetic modification.

“In our opinion, every strategy including stem cells of every origin albeit differen-tiated into insulin-containing cells or whole islets for a mere grafting doom to failure, as the history of the previous studies teaches, due to the fact that the grafted cells or islets will be inevitably de novo attacked.”

Change to “In our opinion, every strategy including stem cells of every origin albeit differen-tiated into insulin-containing cells or whole islets for a mere grafting doom to failure, as the history of the previous studies teaches. This due to the fact that the grafted cells or islets will be inevitably de novo attacked, unless other strategies are undertakes, for instance by encapsulation (doi: 10.1002/adhm.202100965) tor gene editing approaches that protect them from the immune attack (doi: 10.1126/scitranslmed.abn1716).”

  1. Cell therapy and the organoid perspective

  • Lines 354 to 359: references are required. Include: PMID: 36272021, PMID: 36001981,

                  We added suggested references.

                  doi: 10.1016/j.celrep.2022.111238

                  DOI: 10.1007/s12015-022-10391-3

  • Lines 381- 382. Recent publication by Cuesta-Gomez et al has established the critical quality control and release criteria for clinical implementation of stem cell islets (PMID: 36001981). 

“methods to generate cells that meet quality and safety standards for clinical applications still require much further refinement”

Change to: “methods to generate cells that meet quality and safety standards for clinical applications have been recently addressed (doi: 10.1016/j.celrep.2022.111238), despite they are still far from application in clinical setting.”

  • Line 385: ViaCyte and Vertex clinical trials should be mentioned.

                  “But the safety of such therapies is yet to be established in humans.” Change to: “Some recent reports from phase I/II clinical trials showed the feasibility of such treatments in humans, although with a small n (https://doi.org/10.1016/j.xcrm.2021.100466; NCT04786262).”

  • Lines 395 to 410. How could this be implemented into the clinic? Are the authors suggesting isolating this population from human islet donors and expand it to overcome the lack of islet donors?

                  Add the following sentence at the end of the paragraph: “This approach, once validated in humans, could represent an interesting source of human islets.”

  • Lines 411 to 415: are the authors suggesting partial or total removal of the pancreas from the T1D patient during early diagnosis to isolate Procr+ cells? Generation of stem cell islets from peripheral blood would entail a reduced burden to the donor with the same auto-immunity complications. 

Furthermore, it would first need to be determined whether Procr+ progenitors from T1D patients are able to differentiate with the same efficiency as those from healthy controls. 

We softened the conclusion.

“In conclusion, the ideal cell-based therapy of T1D of the future (fig. 1) should move from an early diagnosis (possibly thanks to CEUS) allowing a sufficient isolation of Procr+ progenitors, followed by the generation and expansion of islet organoids, which could be transplanted coupled to an immune-regulatory therapy which will permit the maintenance of pancreatic islets and an effective and long-lasting insulitis reversal.”

Change to:

“In conclusion, one of the proposed approaches for cell-based therapy of T1D in the future (fig. 1) could combine an early diagnosis (possibly thanks to CEUS) with a sufficient isolation of Procr+ progenitors, followed by the generation and expansion of islet organoids, which could be transplanted coupled to an immune-regulatory therapy which will permit the maintenance of pancreatic islets and an effective and long-lasting insulitis reversal. A key issue to resolve would be the maximization of progenitors’ isolation and organoids expansion, such that only a few biopsies would ensure an optimal outcome.”

  • Line 417. Figure legend needs to be expanded to describe the proposed cell-based treatment and resolution of the image must be improved.

                  Proposed cell-based treatment of type 1 diabetes. The proposed approach must combine a CEUS early diagnosis with enough Procr+cell progenitors, combined with endothelial cells derived from IPSCs, to generate and expand islet organoids, that can be grafted in patients undergoing an immune-regolatory therapy, allowing to maintain for long the transplanted pancreatic islets.                   

Round 2

Reviewer 2 Report

I believe the manuscript is ready to be published as is. There is some language/ style that denotes that the authors are not fluent in English but the manuscript is well organized and described and I believe it is ready to be accepted.

Author Response

Thank you for your time in reviewing the manuscript.
We addressed your comments and in particular:
-the whole manuscript has been revised with a native English speaker and made all the amendments. You can find them tracked in red colour within the text;
- the Figure 1 Caption has been revised in order to make it more understandable (please see the modifications in red colour);

-Figure 1 has been changed adding an arrow in order to understand that the immune strategies must be done in patients and the appearance of pancreatic organoids.